# Influence of Sea Level Anomaly on Underwater Gravity Gradient Measurements

**DOI:** 10.3390/s22155758

**Published:** 2022-08-02

**Authors:** Pengfei Xian, Bing Ji, Shaofeng Bian, Bei Liu

**Affiliations:** Department of Navigation, Naval University of Engineering, Wuhan 430033, China; xianpengfei@stu.ouc.edu.cn (P.X.); sfbian@sina.com (S.B.); bqt1900205062@student.cumtb.edu.cn (B.L.)

**Keywords:** disturbing gravity gradient, sea level anomaly, right rectangular prism, forward modeling

## Abstract

Considering the theoretical research needs of gravity gradient detection and navigation, this study uses the right rectangular prism method to calculate the disturbing gravity gradient from sea level anomalies in the range of 5° × 5° in the Kuroshio extension area of the western Pacific with large sea level anomalies. The disturbing gravity gradient is obtained in different directions within a depth of 50 m below the mean sea level based on the principle of the disturbing gravity gradient. The calculation results show that the sea level anomalies at local positions significantly impact the underwater gravity gradient measurements, with the maximum contribution exceeding 10 E and the maximum difference between different locations exceeding 20 E. The change of the sea level anomaly over time also significantly impacts the measurement of the underwater gravity gradient, with the maximum change value exceeding 20 E. The impact will have a corresponding change with the seasonal change of the sea level anomaly. Therefore, the underwater carrier needs to consider the disturbing gravity gradient caused by sea level anomalies when using the gravity gradient for underwater detection and navigation.

## 1. Introduction

Gravity field is an inherent physical property of the earth. Due to differences in seabed topography and crustal density at different locations, the gravity field parameters at different locations in the ocean vary. Passive navigation can be realized using this feature. Gravity gradient denotes the rate of change of gravity in the entire space and is the second derivative of the gravity potential. Gradient anomalies can reflect the details of the field source and exhibit higher resolution than gravity. During detection and navigation using the gravity gradient, a carrier does not actively transmit signals to the outside, thus providing natural advantage for submarines with high concealment.

First, a high-precision and high-resolution gravity gradient reference map needs to be prepared and assembled into the navigation system. The gravity gradiometer is used to measure the gravity gradient data at the carrier location in real time. The measured data are compared with the data of the previously determined gravity gradient reference map. Subsequently, the location of the carrier can be determined by matching and solving according to a certain algorithm, and then, the output results of the inertial navigation can be corrected. The spatial resolution and accuracy of a gravity gradient reference map directly determine the accuracy of navigation and detection. Since the gravity gradiometer has not yet been employed in large-scale engineering applications, the gravity gradient background field needs to be calculated using other models. One possible method is using the Digital Elevation Model (DEM) to forward the gravity gradient [1,2,3,4]. When using DEM data to calculate the gravity gradient, the mean sea level or the geoid, which is approximated by the mean sea level, is used as the datum [5,6,7]. However, certain differences are present between the actual and mean sea levels.

Yan, Z. et al. conducted a quantitative analysis and calculation of the high-frequency wave of the sea surface, and they determined that for the sea state above level 4 (the average wave height is greater than 1.25 m), the fluctuation of the sea waves will cause significant disturbance in the measurement value of the gravity gradient on the underwater carrier [8]. In addition to the high-frequency waves, there are also low-frequency changes in sea level. Sea level denotes the average height of the sea surface over a period of time. It is the level that the ocean can maintain at a certain moment, assuming that no sea surface fluctuation is caused by tides, waves, or other factors. Based on the observation time, the sea level can be divided into daily mean sea level, monthly mean sea level, annual mean sea level, long-term mean sea level, etc., which are affected by weather, climate, or celestial movement and change. Moreover, long-term global trend changes, periodic changes, and large spatial differences can be observed at different locations [9,10,11]. This study aims to determine how the mass anomaly caused by the difference of the sea level height affects the gravity gradient, which could provide a theoretical reference for the practical application of gravity gradient detection and navigation in the future.

## 2. Data and Methods

This study uses daily gridded sea level data from 1993 to the present from global ocean satellite observations that were provided by the Copernicus Marine Environment Monitoring Service (CMEMS) and the Copernicus Climate Change Service (C3S). The dataset provides gridded daily global estimates of sea level anomalies based on satellite altimetry with a spatial resolution of 0.25° × 0.25°, including absolute dynamic terrain, sea level anomaly, absolute geostrophic velocity, and geostrophic velocity anomaly. Moreover, sea level anomalies were calculated relative to the 20-year average reference period (1993–2012). The dataset is based on a satellite constellation with a stable number of altimeters (Figure 1) to ensure long-term stability of the ocean observing system. After the steps of satellite acquisition, preprocessing, input data quality control, multi-task cross-calibration, along-orbit product generation, gridded product generation, and final quality control, estimates of sea level anomalies, absolute dynamic topography and corresponding geostrophic velocities were obtained [12,13,14].

Gravity gradient forward modeling is based on Newton’s law of gravity and utilizes known mass anomalies to calculate the gravity gradient. The methods for the gravity gradient forward modeling from the terrain mainly include right rectangular prism, polyhedron, direct numerical integration, and fast Fourier transform method. Comparison of the various methods showed that the right rectangular prism method affords results with the highest accuracy [2,3,16]; thus, this study uses the right rectangular prism method for calculations.

## 3. Spatial and Temporal Distribution of Sea Level Anomalies

Since the sea level has a significant one-year cycle and the carrier generally does not operate underwater for more than one year, this study chooses 2020 as the representative year. The daily mean sea level anomaly data of 366 days from 1 January 2020 to 31 December 2020 are used to calculate and analyze the impact of sea level height anomaly on the underwater gravity gradient measurements. The spatial distribution of the annual mean sea level anomaly in 2020 (Figure 2) was obtained by time-averaging the 366-day data. The results show that most of the world’s oceans were above the long-term mean sea level in 2020, which is consistent with the current trend of rising sea levels due to global warming. Significant regional differences in the sea level are also observed. Furthermore, the amplitude of the sea level anomaly in the middle of the ocean is small, and positive and negative anomalies with large amplitudes are present in the west boundary current area of the mid latitude ocean and the Antarctic circumpolar current area of the Southern Ocean. Among them, the Kuroshio extension area of the western boundary current in the Pacific Ocean is the most significant, with the highest value of 0.95 m (35.9° N, 145.9° E) and the lowest value of −0.83 m (31.9° N, 136.4° E).

The difference between the maximum and minimum values in the 366 days at each grid point is used to obtain the annual range of global sea level anomalies in 2020 (Figure 3), representing the variation range of the sea level height in 2020. Figure 3 shows that the high-value areas of the annual range are mainly concentrated in the western boundary flow area of the ocean and the Southern Ocean, with amplitudes exceeding 1 m, which indicates that the sea level significantly changes in these areas during the year. The sea area near the southern Cape of Good Hope in Africa (39.4° S, 19.6° E) has an amplitude of 2.11 m, which exhibits the maximum annual range of global sea level anomalies in 2020.

The above analysis shows that the sea level anomaly varies across different regions of the world with a maximum difference of more than 1 m, and the amplitude of changes within a year also differs, with a maximum change of 2 m. Therefore, this study aims to determine whether the existence of sea level anomalies and their variation with the spatiotemporal distribution affect the underwater gravity gradient measurements.

## 4. Simulation Calculation of Gravity Gradient

Considering the large amplitudes of the annual mean and annual range in the Kuroshio extension area in Figure 2 and Figure 3, the right rectangular prism method is used to calculate the disturbing gravity gradient from sea level anomalies in the range of 5° × 5° (34° N–39° N, 152° E–157° E) in the Kuroshio extension area with large sea level anomalies. Moreover, the data of the day with the highest sea level anomaly in the region (8 October 2020) is selected from the 366-day data for subsequent calculation (Figure 4 and Table 1).

Due to the small longitudinal and latitudinal span of the study area, the calculation area is regarded as a plane, and a local Cartesian coordinate system is established on the mean sea level with the x, y, and z axes pointing toward the east, north, and upward directions (Figure 5). The projection of the computation point on the mean sea level is set as the origin, which will change the horizontal coordinates of the computation point P to 0, subsequently simplifying the calculation. Ignoring the influence of the centrifugal force generated by the earth’s rotation, the sea level anomaly data are divided into rectangular prisms based on the longitude and latitude grid, whose length and width are determined as the average length and width of all grids and the upper and lower surfaces of the prism are determined as the sea level anomaly surface and mean sea level, respectively. Then, the horizontal position of the computation point is at the center of each grid (Figure 6).

To simplify the formula, using $x_{1}$, $x_{2}$, and $x_{3}$ to represent *x*, *y*, and *z*, respectively, we obtain the gravity gradient generated by the *i*th prism at the computation point P:(1)$$\Gamma_{jk}^{i}=G\int_{0}^{h_{i}}\int_{b_{i}-\frac{\Delta x_{2}}{2}}^{b_{i}+\frac{\Delta x_{2}}{2}}\int_{a_{i}-\frac{\Delta x_{1}}{2}}^{a_{i}+\frac{\Delta x_{1}}{2}}\frac{\partial^{2}}{\partial x_{j}\partial x_{k}}\frac{1}{r}\rho_{i}dx_{1}^{\prime }dx_{2}^{\prime }dx_{3}^{\prime }$$

Here, G denotes Newton’s gravitation constant, which is taken as 6.67 × 10^−11^ m^3^·kg^−1^·s^−2^; $\rho_{i}$ is the seawater density of the *i*th prism, which is assumed to be a constant value of 1025 kg·m^−3^; $h_{i}$ is the sea level anomaly value of the *i*th prism; and ($a_{i}$, $b_{i}$) is the horizontal coordinate of the center point of the *i*th prism. Additionally, $\Delta x_{1}$ and $\Delta x_{2}$ are the average length and width, respectively, of each grid in the study area, which are calculated to be 22,371.490 and 27,742.137 m. Furthermore, $r=\sqrt{{\left(x_{1}-x_{1}^{\prime }\right)}^{2}+{\left(x_{2}-x_{2}^{\prime }\right)}^{2}+{\left(x_{3}-x_{3}^{\prime }\right)}^{2}}$ is the distance between the computation point P ($x_{1}$, $x_{2}$, $x_{3}$) and element point P’ ($x_{1}^{\prime }$, $x_{2}^{\prime }$, $x_{3}^{\prime }$), *j* and *k* are the subscripts of *x*, which can be taken as 1, 2, and 3, representing the *x*, *y*, and *z* directions, respectively.

Substitute *x*_1_ = 0, *x*_2_ = 0, and *x*_3_ = *Z* into Equation (1), place *ρ* before the integral sign, and replace $x_{1}^{\prime }$, $x_{2}^{\prime }$ with *u*, *v* to represent the horizontal coordinates of the element point. Next, integrate Equation (1) in the vertical direction to obtain six components:(2)$$\begin{array}{c} \Gamma_{xx}^{i}=G\rho \int_{b_{i}-\frac{\Delta x_{2}}{2}}^{b_{i}+\frac{\Delta x_{2}}{2}}\int_{a_{i}-\frac{\Delta x_{1}}{2}}^{a_{i}+\frac{\Delta x_{1}}{2}}\frac{1}{{\left(u^{2}+v^{2}\right)}^{2}}(\frac{Z\left(2u^{4}+u^{2}v^{2}-v^{4}+\left(u^{2}-v^{2}\right)Z^{2}\right)}{{\left(u^{2}+v^{2}+Z^{2}\right)}^{3/2}}\\ +\frac{\left(h_{i}-Z\right)\left(2u^{4}+u^{2}v^{2}-v^{4}+\left(u^{2}-v^{2}\right){\left(h_{i}-Z\right)}^{2}\right)}{{\left(u^{2}+v^{2}+{\left(h_{i}-Z\right)}^{2}\right)}^{3/2}})dudv \end{array}$$
(3)$$\begin{array}{c} \Gamma_{yy}^{i}=G\rho \int_{b_{i}-\frac{\Delta x_{2}}{2}}^{b_{i}+\frac{\Delta x_{2}}{2}}\int_{a_{i}-\frac{\Delta x_{1}}{2}}^{a_{i}+\frac{\Delta x_{1}}{2}}\frac{1}{{\left(u^{2}+v^{2}\right)}^{2}}(\frac{Z\left(2v^{4}+u^{2}v^{2}-u^{4}+\left(v^{2}-u^{2}\right)Z^{2}\right)}{{\left(u^{2}+v^{2}+Z^{2}\right)}^{3/2}}\\ +\frac{\left(h_{i}-Z\right)\left(2v^{4}+u^{2}v^{2}-u^{4}+\left(v^{2}-u^{2}\right){\left(h_{i}-Z\right)}^{2}\right)}{{\left(u^{2}+v^{2}+{\left(h_{i}-Z\right)}^{2}\right)}^{3/2}})dudv \end{array}$$
(4)$$\Gamma_{zz}^{i}=-G\rho \int_{b_{i}-\frac{\Delta x_{2}}{2}}^{b_{i}+\frac{\Delta x_{2}}{2}}\int_{a_{i}-\frac{\Delta x_{1}}{2}}^{a_{i}+\frac{\Delta x_{1}}{2}}\frac{Z}{{\left(u^{2}+v^{2}+Z^{2}\right)}^{3/2}}+\frac{h_{i}-Z}{{\left(u^{2}+v^{2}+{\left(h_{i}-Z\right)}^{2}\right)}^{3/2}}dudv$$
(5)$$\begin{array}{c} \Gamma_{xy}^{i}=G\rho \int_{b_{i}-\frac{\Delta x_{2}}{2}}^{b_{i}+\frac{\Delta x_{2}}{2}}\int_{a_{i}-\frac{\Delta x_{1}}{2}}^{a_{i}+\frac{\Delta x_{1}}{2}}\frac{1}{{\left(u^{2}+v^{2}\right)}^{2}}(\frac{Z\left(uv\left(3u^{2}+3v^{2}+2Z^{2}\right)\right)}{{\left(u^{2}+v^{2}+Z^{2}\right)}^{3/2}}\\ +\frac{\left(h_{i}-Z\right)\left(u\;v\left(3u^{2}+3v^{2}+2{\left(h_{i}-Z\right)}^{2}\right)\right)}{{\left(u^{2}+v^{2}+{\left(h_{i}-Z\right)}^{2}\right)}^{3/2}})dudv \end{array}$$
(6)$$\Gamma_{xz}^{i}=G\rho \int_{b_{i}-\frac{\Delta x_{2}}{2}}^{b_{i}+\frac{\Delta x_{2}}{2}}\int_{a_{i}-\frac{\Delta x_{1}}{2}}^{a_{i}+\frac{\Delta x_{1}}{2}}u\left(\frac{1}{{\left(u^{2}+v^{2}+Z^{2}\right)}^{3/2}}-\frac{1}{{\left(u^{2}+v^{2}+{\left(h_{i}-Z\right)}^{2}\right)}^{3/2}}\right)dudv$$
(7)$$\Gamma_{yz}^{i}=G\rho \int_{b_{i}-\frac{\Delta x_{2}}{2}}^{b_{i}+\frac{\Delta x_{2}}{2}}\int_{a_{i}-\frac{\Delta x_{1}}{2}}^{a_{i}+\frac{\Delta x_{1}}{2}}v\left(\frac{1}{{\left(u^{2}+v^{2}+Z^{2}\right)}^{3/2}}-\frac{1}{{\left(u^{2}+v^{2}+{\left(h_{i}-Z\right)}^{2}\right)}^{3/2}}\right)dudv$$

*Z* denotes the depth from the mean sea level, which is a negative value. Selecting a certain depth and substituting it into the formula can yield the components of the gravity gradient. When the selected depth *Z* = 0, Equations (2)–(7) can be simplified:(8)$$\Gamma_{xx}^{i}=G\rho \int_{b_{i}-\frac{\Delta x_{2}}{2}}^{b_{i}+\frac{\Delta x_{2}}{2}}\int_{a_{i}-\frac{\Delta x_{1}}{2}}^{a_{i}+\frac{\Delta x_{1}}{2}}\frac{h_{i}\left(2u^{4}+u^{2}v^{2}-v^{4}+h_{i}{}^{2}\left(u^{2}-v^{2}\right)\right)}{{\left(u^{2}+v^{2}\right)}^{2}{\left(u^{2}+v^{2}+h_{i}{}^{2}\right)}^{3/2}}dudv$$
(9)$$\Gamma_{yy}^{i}=G\rho \int_{b_{i}-\frac{\Delta x_{2}}{2}}^{b_{i}+\frac{\Delta x_{2}}{2}}\int_{a_{i}-\frac{\Delta x_{1}}{2}}^{a_{i}+\frac{\Delta x_{1}}{2}}\frac{h_{i}\left(2v^{4}+u^{2}v^{2}-u^{4}+h_{i}{}^{2}\left(v^{2}-u^{2}\right)\right)}{{\left(u^{2}+v^{2}\right)}^{2}{\left(u^{2}+v^{2}+h_{i}{}^{2}\right)}^{3/2}}dudv$$
(10)$$\Gamma_{zz}^{i}=-G\rho \int_{b_{i}-\frac{\Delta x_{2}}{2}}^{b_{i}+\frac{\Delta x_{2}}{2}}\int_{a_{i}-\frac{\Delta x_{1}}{2}}^{a_{i}+\frac{\Delta x_{1}}{2}}\frac{h_{i}}{{\left(u^{2}+v^{2}+h_{i}{}^{2}\right)}^{3/2}}dudv$$
(11)$$\Gamma_{xy}^{i}=G\rho \int_{b_{i}-\frac{\Delta x_{2}}{2}}^{b_{i}+\frac{\Delta x_{2}}{2}}\int_{a_{i}-\frac{\Delta x_{1}}{2}}^{a_{i}+\frac{\Delta x_{1}}{2}}\frac{h_{i}\;u\;v\left(3u^{2}+3v^{2}+2\;h_{i}{}^{2}\right)}{{\left(u^{2}+v^{2}\right)}^{2}{\left(u^{2}+v^{2}+h_{i}{}^{2}\right)}^{3/2}})dudv$$
(12)$$\Gamma_{xz}^{i}=G\rho \int_{b_{i}-\frac{\Delta x_{2}}{2}}^{b_{i}+\frac{\Delta x_{2}}{2}}\int_{a_{i}-\frac{\Delta x_{1}}{2}}^{a_{i}+\frac{\Delta x_{1}}{2}}u\left(\frac{1}{{\left(u^{2}+v^{2}\right)}^{3/2}}-\frac{1}{{\left(u^{2}+v^{2}+h_{i}{}^{2}\right)}^{3/2}}\right)dudv$$
(13)$$\Gamma_{yz}^{i}=G\rho \int_{b_{i}-\frac{\Delta x_{2}}{2}}^{b_{i}+\frac{\Delta x_{2}}{2}}\int_{a_{i}-\frac{\Delta x_{1}}{2}}^{a_{i}+\frac{\Delta x_{1}}{2}}v\left(\frac{1}{{\left(u^{2}+v^{2}\right)}^{3/2}}-\frac{1}{{\left(u^{2}+v^{2}+h_{i}{}^{2}\right)}^{3/2}}\right)dudv$$

Each gravity gradient component of prism I at point P can be calculated using Equations (2)–(7), and its value at point P can be obtained by adding the disturbing gravity gradient values of all prisms in the calculation area at point P. Then, the disturbing gravity gradient caused by the sea level anomaly in the study area can be obtained through point by point calculation of the computation point of all grid centers [17].

The horizontal coordinate of the center point of the *i*th prism ($a_{i},\;b_{i}$) in Equation (1) should cover all mass elements since all mass elements will affect the gravity gradient at the computation point; however, this is difficult to achieve in practical calculations. Equation (1) shows that the gravity gradient rapidly decreases with increasing distance, denoting that the contribution of distant mass elements to the gradient value is very small. Therefore, the range of the calculation area needs to be determined to consider the balance between the calculation efficiency and error. Jekeli, C. developed a systematic algorithm method to determine the range of terrain data [18]. Wu, L. took the relative error of 1% as the truncation distance to calculate the gravity gradient effect [7]. We adopt Wu’s method and select the locations corresponding to the highest value, the lowest value, and any other value in the study area as test points, and then, gradually increase the side length of the integration area to calculate the vertical disturbing gravity gradient at different depths below the mean sea level caused by the sea level anomalies (Figure 7 and Table 2, Table 3 and Table 4). Figure 7 shows that with increasing integration area, the vertical disturbing gravity gradient tends to be stable and the change decreases. When the gradient tends to stabilize, the side length of the integral region increases as the observation depth increases. To achieve a relative error of less than 1%, the side length of the three test points at 10 m below the mean sea level in the integration area needs to reach 24 m, which is 2.4 times the observation point depth. The side length of the three test points at 30 m below the mean sea level needs to reach 68 m, which is 2.3 times the observation point depth. The side length of the three test points at 50 m below the mean sea level needs to reach 108 m, which is 2.2 times the observation point depth. Furthermore, the gravity gradient forward modeling result is related to the integration range, and the selection of the integration range is related to the calculation height (depth). Therefore, to obtain relatively accurate forward modeling results, different integration ranges need to be considered when calculating at different heights (depths).

## 5. Experimental Results and Analysis

When the relative error is less than 1%, 24, 68, and 108 m can be taken as the side lengths of the integration area to calculate the disturbing gravity gradient at 10, 30, and 50 m below the mean sea level, respectively. Since the average grid side length of the sea level anomaly data is about 20 km, which is significantly larger than the side length of the integration area, the gravity gradient value of each grid is only calculated from the mass anomaly in the central area of the grid, i.e., only one prism is calculated at each grid. Only the three components of *Γ_xx_*, *Γ_yy_*, and *Γ_zz_* are calculated and analyzed below since the *Γ_xy_*, *Γ_xz_*, and *Γ_yz_* of a single prism on the vertical line where the midpoint is located are zero [19].

Figure 8 shows that the sum of *Γ_xx_*, *Γ_yy_*, and *Γ_zz_* is close to zero, satisfying the Laplace equation, and thus, the calculation process can be considered to be correct. The chart shows that the sea level anomaly considerably impacts the disturbing gravity gradient value at 10 m below the mean sea level, up to 16 E, and the difference between the disturbing gravity gradient at different locations can reach 22 E. At 30 and 50 m below the mean sea level, the spatial distribution of *Γ_xx_*, *Γ_yy_*, and *Γ_zz_* is similar to the distribution at 10 m below the mean sea level, but the magnitude is lower; thus, only the statistics of the calculation results are listed. Table 5 shows that the disturbing gravity gradient caused by the sea level anomaly significantly decreases with increasing depth. The vertical disturbing gravity gradient can still reach 3 E at 50 m below the mean sea level, and the maximum difference of the disturbing gravity gradient at different locations can reach 4 E. Therefore, based on the measurement accuracy of the gravity gradiometer 1 E [20,21,22,23,24,25], the local sea level anomaly will considerably affect the underwater gravity gradient measurements at the depth of 50 m below the mean sea level.

The above results show that in addition to the spatial differences, the sea level anomalies will change with time. The annual range of the sea level anomalies in this area (Figure 9) is used to forward calculate the disturbing gravity gradient. The maximum and minimum values of the sea level anomaly are used as the upper and lower surfaces of the prism, respectively. The calculation process is the same as above. We obtain the change value of the disturbing gravity gradient caused by sea level anomaly within one year (Figure 10). Figure 10 and Table 6 show that if the carrier measured the gravity gradient for one year in 2020 at 10 m below the mean sea level in the study area, the variation of the vertical disturbing gravity gradient caused by the sea level anomaly variation with time could reach a maximum of 24 E and minimum of 3 E, and the variation of the horizontal disturbing gravity gradient could reach a maximum of 10 E and minimum of 1 E. At 50 m below the mean sea level, the maximum variation of the vertical disturbing gravity gradient is more than 4 E, while the minimum variation is less than 1 E. We select the location of the maximum annual range in the study area (36.6° N, 154.4° E) and calculate the disturbing gravity gradient for one year in 2020 (Figure 11). We find that the disturbing gravity gradient will be larger in summer and autumn compared to the other seasons, which is closely related to the seasonal changes in sea level anomalies. Therefore, within a depth of 50 m below the mean sea level, the change of the local sea level anomaly with time will have different effects on the underwater gravity gradient measurements, and this effect will change according to the seasonal changes of the sea level anomalies.

## 6. Conclusions

Herein, we processed the daily grid data of sea level observed by global ocean satellites and quantitatively calculated the disturbing gravity gradient caused by sea level anomalies using the right rectangular prism method. The results show the following:(1)The gravity gradient forward modeling results are related to the integration range, and the selection of the integration range is related to the calculation height (depth). Therefore, to obtain relatively accurate forward modeling results, different integration ranges need to be considered when calculating at different heights (depths).(2)Based on the measurement accuracy of the gravity gradiometer 1 E, within 50 m below the mean sea level, sea level anomalies at local positions will significantly affect the underwater gravity gradient measurements, with a maximum contribution exceeding 10 E and the maximum difference between different locations exceeding 20 E. Moreover, the change of the sea level anomalies with time will significantly impact the underwater gravity gradient measurements, with the maximum change value exceeding 20 E, and the impact will accordingly change with the seasonal change of the sea level anomalies. Therefore, underwater carriers need to consider the disturbing gravity gradient caused by sea level anomalies when using gravity gradient for underwater detection and navigation.

## Figures and Tables

**Figure 1 sensors-22-05758-f001:**
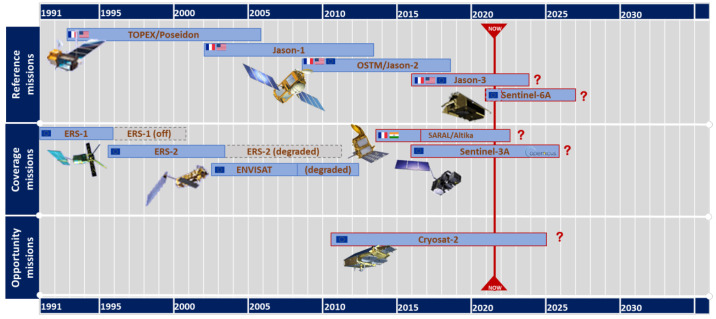
Complete altimetry satellite constellation used in the C3S sea level product [15].

**Figure 2 sensors-22-05758-f002:**
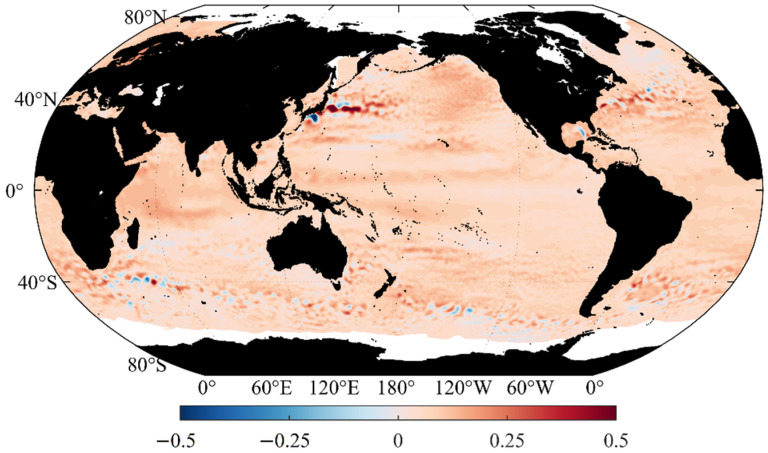
Global annual mean sea level anomaly in 2020 (unit: m).

**Figure 3 sensors-22-05758-f003:**
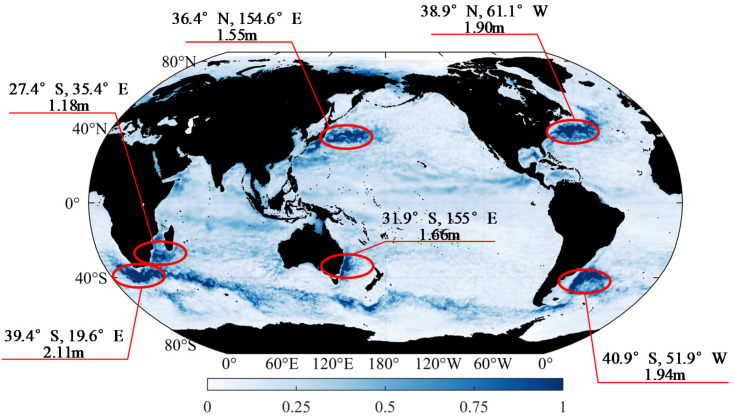
Annual range of global sea level anomalies in 2020 (unit: m).

**Figure 4 sensors-22-05758-f004:**
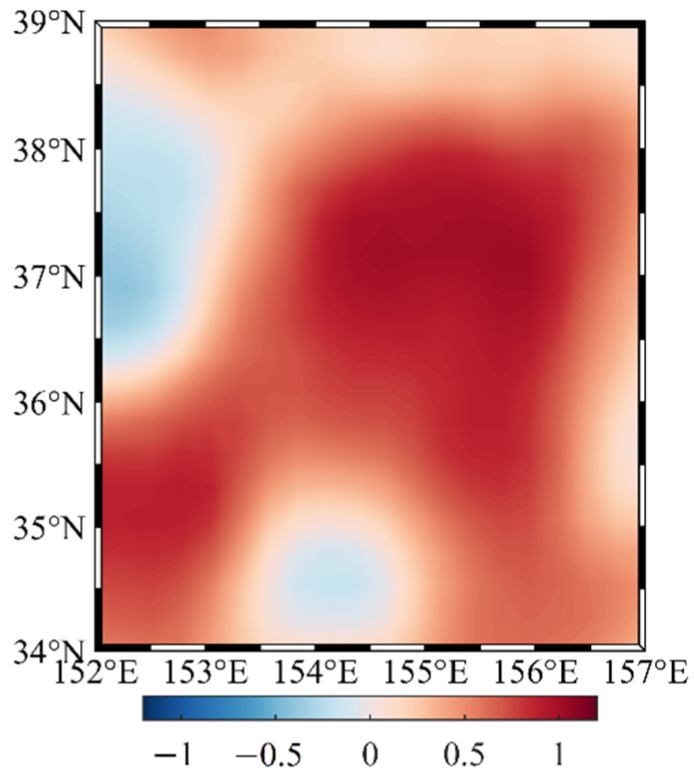
Annual range of the global sea level anomalies in 2020 (unit: m).

**Figure 5 sensors-22-05758-f005:**
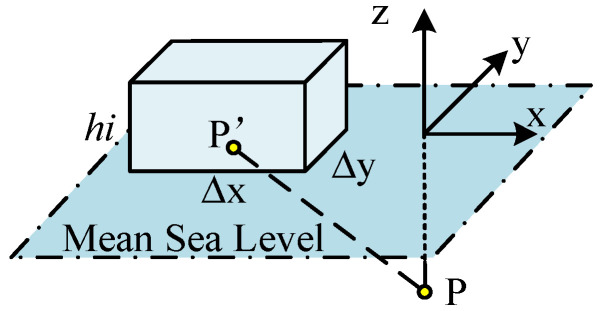
Right rectangular prism.

**Figure 6 sensors-22-05758-f006:**
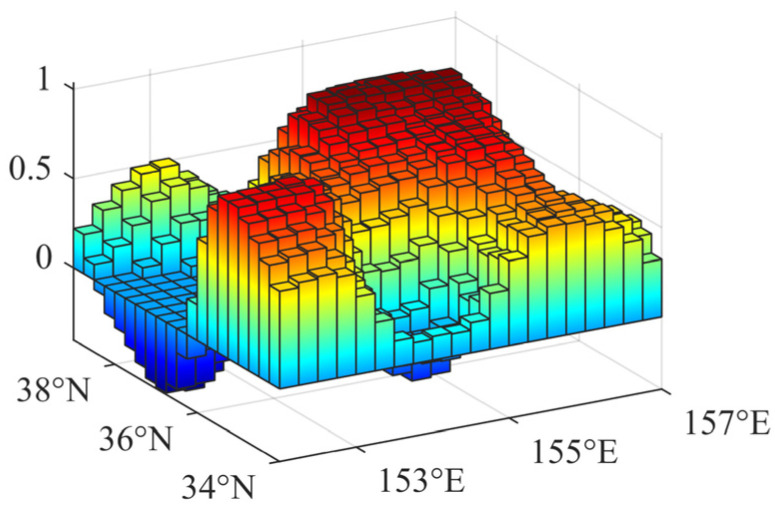
Block representation of the sea level anomalies (unit: m).

**Figure 7 sensors-22-05758-f007:**
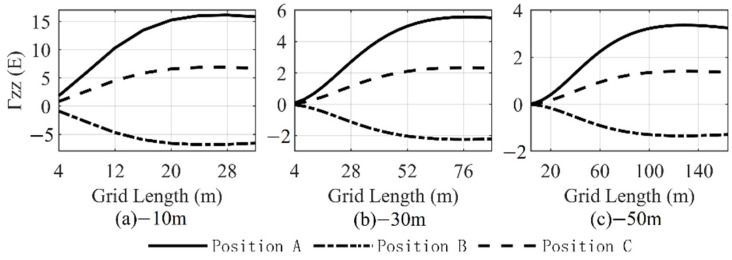
Relation between gradient and integration range.

**Figure 8 sensors-22-05758-f008:**
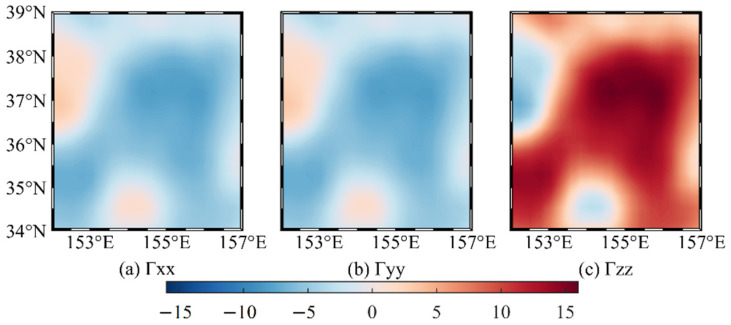
Disturbing gravity gradient at 10 m below mean sea level (unit: E).

**Figure 9 sensors-22-05758-f009:**
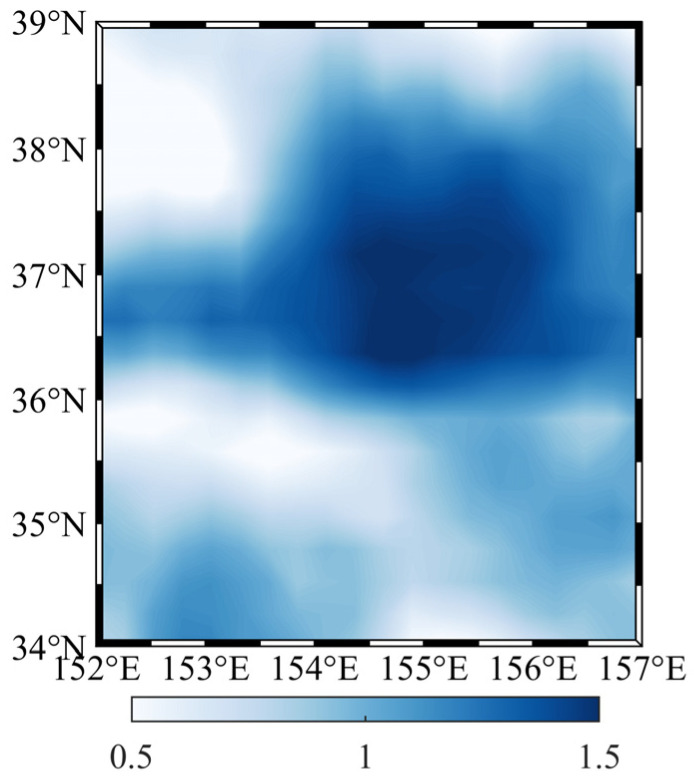
Annual range of sea level anomalies in the study area (unit: m).

**Figure 10 sensors-22-05758-f010:**
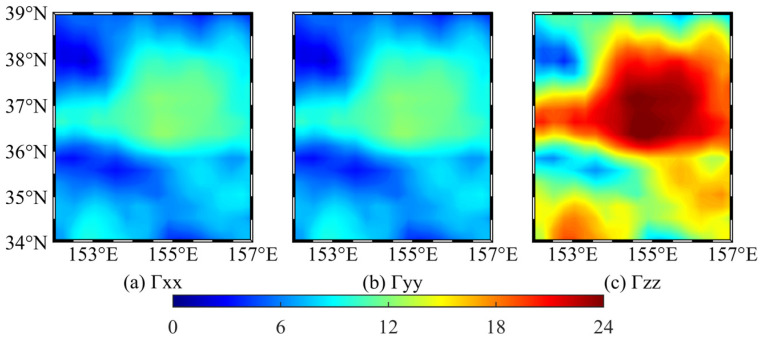
Annual range of the disturbing gravity gradient at 10 m below the mean sea level (*Γ_xx_* and *Γ_yy_* are both negative, and the absolute values shown in (**a**,**b**). *Γ_zz_* is positive in (**c**). (unit: E).

**Figure 11 sensors-22-05758-f011:**
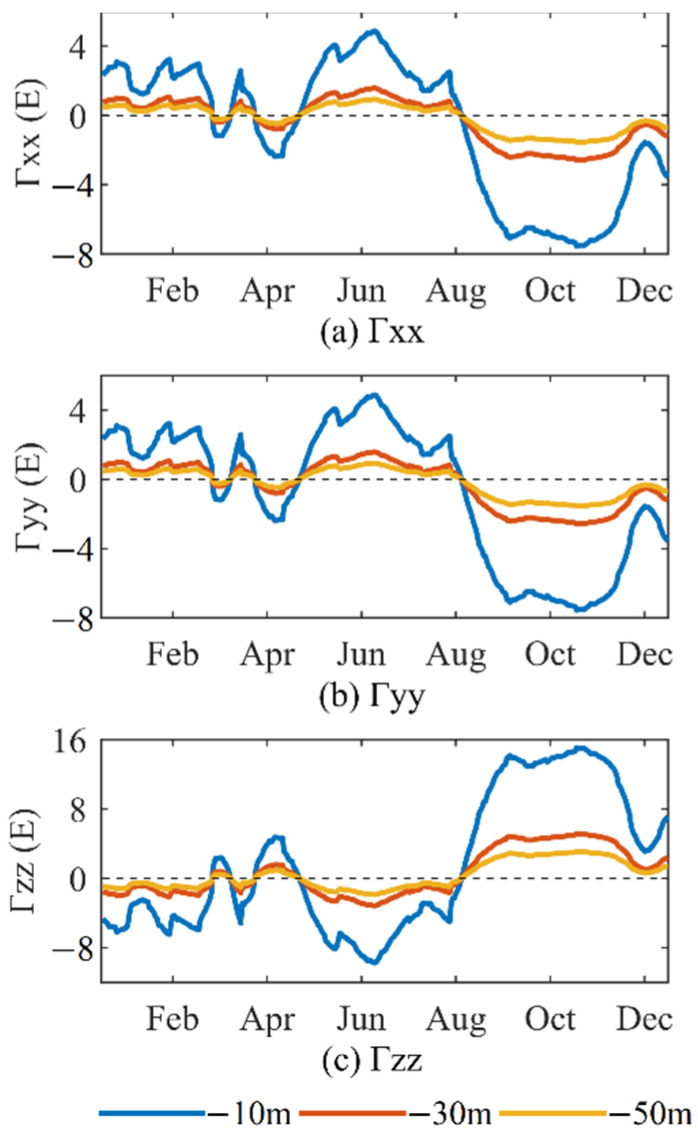
Variation of the disturbing gravity gradient with time at the location of the maximum annual range.

**Table 1 sensors-22-05758-t001:** Parameters of the sea level anomaly data in the study area.

Lat. and Lon.	Resolution	Number of Grids	Max (m)	Min (m)	STD (m)	Mean (m)
34° N–39° N 152° E–157° E	0.25° × 0.25°	400	1.0347	−0.4070	0.3537	0.4768

**Table 2 sensors-22-05758-t002:** Variation of gradient with integration range at 10 m below mean sea level.

Side Length of Integration Area (m)	Vertical Gravity Gradient at 10 m below Mean Sea Level (E)		
	Position A	Position B	Position C
4	1.821	−0.874	0.819
8	6.008	−2.813	2.675
12	10.305	−4.687	4.537
16	13.446	−5.952	5.858
20	15.258	−6.603	6.588
**24**	**16.038**	**−6.818**	**6.874**
28	16.140	−6.765	6.878
32	15.838	−6.566	6.719

**Table 3 sensors-22-05758-t003:** Variation of gradient with integration range at 30 m below mean sea level.

Side Length of Integration Area (m)	Vertical Gravity Gradient at 30 m below Mean Sea Level (E)		
	Position A	Position B	Position C
44	4.441	−1.823	1.876
48	4.738	−1.939	1.999
52	4.981	−2.033	2.099
56	5.174	−2.106	2.178
60	5.322	−2.161	2.238
64	5.430	−2.200	2.281
**68**	**5.503**	**−2.226**	**2.310**
72	5.547	−2.239	2.327

**Table 4 sensors-22-05758-t004:** Variation of gradient with integration range at 50 m below mean sea level.

Side Length of Integration Area (m)	Vertical Gravity Gradient at 50 m below Mean Sea Level (E)		
	Position A	Position B	Position C
84	2.966	−1.194	1.242
88	3.045	−1.224	1.275
92	3.113	−1.251	1.303
96	3.172	−1.273	1.327
100	3.220	−1.292	1.347
104	3.262	−1.307	1.364
**108**	**3.295**	**−1.319**	**1.377**
112	3.320	−1.328	1.387

**Table 5 sensors-22-05758-t005:** Statistics of the disturbing gravity gradient at different depths below the mean sea level (unit: E).

Depths	Statistics	Max	Min	Mean	STD	Range
10 m	*Γ_xx_*	3.409	−8.019	−3.754	2.777	11.428
	*Γ_yy_*	3.409	−8.019	−3.754	2.777	11.428
	*Γ_zz_*	16.038	−6.817	7.509	5.554	22.855
30 m	*Γ_xx_*	1.113	−2.751	−1.275	0.945	3.864
	*Γ_yy_*	1.113	−2.752	−1.275	0.945	3.865
	*Γ_zz_*	5.503	−2.226	2.550	1.890	7.729
50 m	*Γ_xx_*	0.659	−1.646	−0.761	0.564	2.305
	*Γ_yy_*	0.660	−1.648	−0.762	0.565	2.308
	*Γ_zz_*	3.295	−1.319	1.524	1.129	4.614

**Table 6 sensors-22-05758-t006:** Statistics of the annual range of the disturbing gravity gradient at different depths below the mean sea level (unit: E).

Depths	Statistics	Max	Min	Mean	STD	Range
10 m	*Γ_xx_*	−1.941	−12.428	−7.678	2.401	10.486
	*Γ_yy_*	−1.942	−12.428	−7.679	2.401	10.487
	*Γ_zz_*	24.856	3.883	15.357	4.803	20.973
30 m	*Γ_xx_*	−0.637	−4.166	−2.591	0.807	3.529
	*Γ_yy_*	−0.637	−4.167	−2.591	0.807	3.530
	*Γ_zz_*	8.333	1.274	5.182	1.614	7.059
50 m	*Γ_xx_*	−0.378	−2.481	−1.545	0.481	2.104
	*Γ_yy_*	−0.378	−2.484	−1.547	0.482	2.106
	*Γ_zz_*	4.966	0.756	3.092	0.963	4.210

## Data Availability

Publicly available datasets were analyzed in this study. This data can be found here: https://cds.climate.copernicus.eu/cdsapp#!/dataset/satellite-sea-level-global?tab=overview (accessed on 22 October 2021).

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
