# Peer review of "Influence of Sea Level Anomaly on Underwater Gravity Gradient Measurements"

_sensors, 2022, doi:10.3390/s22155758_

Round 1
Reviewer 1 Report
The authors have chosen a very good topic to study. The daily sea level changes relative to the mean sea level constantly generate mass anomalies and hence alter the gravity field in daily bases. The GRACE and its follow on mission cannot efficiently handle this. Apparently, the authors' trailblazing research shed lights on this topic and will be very useful to the future study as well. Hence, I recommend acceptance of this study after some minor revision.
The only thing that bothers me is using only one small prism in each cell...
Can the authors do a global integration to verify the convergence in Fig. 7?
Reviewer 2 Report
This paper shows the impacts of sea level anomaly and its temporal variations on the subsurface gravity gradients, usually neglected by present studies. This topic has broad interest in the geodesy and oceanography communities. Results of this manuscript showed that the temporal variations of the sea level could generate about 20 Eötvös of gravity gradient variations at 10 m depth on the gravity gradient reference maps, which is measured prior to the navigation and thus assumed invariant. Considering that commercial gravity gradiometer (Air-FTG) has reported an accuracy of 5–7 Eötvös in 2013, the impacts of sea level variation need to be considered in the submarine navigation. Therefore, this manuscript is acceptable for publication in Sensors. The manuscript is well written and only needs minor modifications.
The suggested minor revisions are listed below.
Line 48: Please add specific numbers or descriptions to help the readers understand what is the sea state above level 4.
Lines 75-78: “Satellite altimetry has …… geoid, gravity anomalies, etc.” This sentence is irrelevant to the topic of this manuscript, right? It is more helpful to tell the readers the accuracy of the sea level products.
Line 84: “…… method.” Need references, e.g.,
Yang, J., Jekeli, C., & Liu, L. (2018). Seafloor topography estimation from gravity gradients using simulated annealing. Journal of Geophysical Research: Solid Earth, 123(8), 6958-6975. https://doi.org/10.1029/2018JB015883
Zhu, L., & Jekeli, C. (2009). Gravity gradient modeling using gravity and DEM. Journal of Geodesy, 83(6), 557-567. https://doi.org/10.1007/s00190-008-0273-2
Line 129: Is it “Annual range of the sea level anomalies ……”?
Line 157: The gravity gradients for the right rectangular prism has been discussed by Nagy et al., right?
Nagy, D., Papp, G., & Benedek, J. (2000). The gravitational potential and its derivatives for the prism. Journal of Geodesy, 74(7-8), 552-560. https://doi.org/10.1007/s001900000116
Line 169: There is a way to estimate the variance of the truncation error and resolution error based on the power spectral density of the topography. It can be used to determine the extent of mass elements needed to maintain a truncation error below a desired level.
Jekeli, C. (2013). Extent and resolution requirements for the residual terrain effect in gravity gradiometry. Geophysical Journal International, 195(1), 211-221. https://doi.org/10.1093/gji/ggt246
